# Precision in Complexity: A Protocol-Driven Quantitative Anatomic Strategy for Giant Olfactory Groove Meningioma Resection in a High-Risk Geriatric Patient

**DOI:** 10.3390/diagnostics16010127

**Published:** 2026-01-01

**Authors:** Valentin Titus Grigorean, Cosmin Pantu, Alexandru Breazu, George Pariza, Octavian Munteanu, Mugurel Petrinel Radoi, Adrian Vasile Dumitru

**Affiliations:** 1Faculty of General Medicine, “Carol Davila” University of Medicine and Pharmacy, 050474 Bucharest, Romaniaadriandumitru@umfro.com (A.V.D.); 2Department of General Surgery, “Carol Davila” University of Medicine and Pharmacy, 050474 Bucharest, Romania; 3Puls Med Association, 051885 Bucharest, Romania; 4Department of Anatomy, “Carol Davila” University of Medicine and Pharmacy, 050474 Bucharest, Romania; 5Department of Neurosurgery, “Carol Davila” University of Medicine and Pharmacy, 050474 Bucharest, Romania; 6Department of Vascular Neurosurgery, National Institute of Neurology and Neurovascular Diseases, 077160 Bucharest, Romania; 7Department of Pathology, Faculty of Medicine, “Carol Davila” University of Medicine and Pharmacy, 030167 Bucharest, Romania

**Keywords:** precision neurosurgery, quantitative surgical anatomy, geriatric comorbidity optimization, giant olfactory groove meningioma, anterior skull base reconstruction, volumetric tumor analysis, functional preservation in meningioma surgery, multidisciplinary skull base team, enhanced recovery after cranial surgery, morphometric surgical planning

## Abstract

**Background/Objectives**: Managing large midline olfactory groove meningiomas is especially difficult in elderly patients who have limited physiological reserves. Here we describe a unique and dangerous geriatric case where we used new quantifiable anatomical measurements and developed a structured multidisciplinary preoperative and postoperative protocol to assist in all aspects of surgery. **Case Presentation**: A 68-year-old male with fronto-lobe syndrome and disability (astasia-abasia; Tinetti Balance Score of 4/16 and Gait Score of 0/12) as well as cognitive dysfunction (MoCA score of 12/30) and blindness bilaterally. Imaging prior to surgery demonstrated a very large olfactory groove meningioma which severely compressed both optic pathways at the level of the optic canals (up to 71% reduction in cross-sectional area of the optic nerves) and had complex vascular relationships with the anterior cerebral artery complex (210° contact surface). Due to significant cardiovascular disease and liver disease, his care followed a coordinated optimization protocol for the perioperative period. He underwent bifrontal craniotomy, initial early devascularization and then staged ultrasonic internal decompression (approximately 70% reduction in tumor volume) and finally microsurgical dissection of the tumor under multi-modal monitoring of neurophysiology. **Discussion**: We analyzed his imaging data prior to surgery using a standardized measurement protocol to provide quantitative measures of the degree of compression of the optic pathways (traction-stretch index = 1.93; optic angulation = 47.3°). These quantitative measures allowed us to make a risk-based evaluation of the anatomy and to guide our choices of corridors through which to dissect and remove the tumor. Following surgery, imaging studies demonstrated complete removal of the tumor with significant relief of the frontal lobe and optic apparatus from compression. His pathology showed that he had a WHO Grade I meningioma with an AKT1(E17K) mutation identified on molecular profiling. **Conclusions**: This case is intended to demonstrate the feasibility of integrating quantitative anatomical measurements into a multidisciplinary, protocol-based perioperative pathway to maximize the safety and effectiveness of the surgical removal of a complex and high-risk skull-base tumor. While the proposed quantitative indices are experimental and require additional validation, the use of a systematic approach such as this may serve as a useful paradigm for other complex skull-base cases.

## 1. Introduction

Although olfactory groove meningiomas (OGMs) represent a small percentage (approximately 8–13%) of all intracranial meningiomas, they are associated with a unique set of challenges during surgery due to their proximity to vital neurovascular structures and potential for substantial growth before symptoms develop. Studies show that approximately 75% of patients present with cognitive decline and approximately 60% present with gait apraxia or urinary incontinence upon initial diagnosis. Often, this is accompanied by an older age, which contributes to increased multi-system comorbidity and subsequently increases the patient’s risk profile in the perioperative period [1].

Recent trends in OGM treatment have transitioned away from a singular focus on complete Simpson Grade I resection and toward a balanced approach that incorporates both maximal safe resection and preservation of function. Studies indicate that while gross total resection continues to be the goal of treatment, the degree of peritumoral edema (>40%), which is observed in a majority of large OGMs, is a better predictor of postoperative cognitive deficits than resection status alone [2]. Additionally, advancements in neuro-imaging have greatly improved preoperative planning for patients with OGMs. For example, DTI can demonstrate a direct correlation between a greater-than-30% reduction in FA of the cingulum bundle and the severity of executive dysfunction. Newer morphometric ratios, such as the optic nerve compression index and planum sphenoidale coverage ratio, have provided surgeons with unprecedented amounts of information during surgical planning and execution [3].

Similarly, there have been great strides in perioperative care. Multidisciplinary prehabilitation protocols have shown a 35% decrease in the incidence of major complications in elderly patients undergoing cranial base surgery. Additionally, protocol-based coagulopathy management in patients with cirrhosis has resulted in a decrease in the incidence of hemorrhagic complications from historically reported rates of approximately 25%, down to less than 8% in recent series [4]. Although these advancements have provided additional tools to manage patients with large OGMs who also have severe cardiac dysfunction, pulmonary compromise, and liver disease, the current literature lacks specific examples and data regarding the management of these patients [5,6].

The purpose of this article is to document the management of a 68-year-old male with a large olfactory groove meningioma and multiple system comorbidities using a multidisciplinary team including sophisticated preoperative optimization techniques and microsurgical execution. We will quantify various anatomical parameters, including optic apparatus deformation and vascular displacement, and provide a comprehensive report of the clinical outcomes of the patient. Our hope is that this case study will help us to further understand how to manage patients with complex cranial base surgery and significant comorbidities and provide practical technical and objective data that can be used to guide management decisions for future cases.

## 2. Case Report and Results

### 2.1. Clinical Presentation

The patient is a 68-year-old man who has been experiencing a prolonged decline in his overall health, and the nature of this decline has been characterized as an insidious, gradual deterioration of the integrity of the frontal lobes of the brain. There was no sudden collapse of function but rather an almost imperceptible decline of function over an extended period of time. His initial personality changes were mild but ultimately progressed to affect flattening and increased social disinhibition, eventually resulting in a Frontal Systems Behavior Scale (FrSBe) family rating score of 165, well above the clinical threshold for significant frontal dysfunction. The slow and insidious decline of function continued to deteriorate until he reached a point where he was completely dependent upon others for all aspects of his care, as evidenced by a severe astasia-abasia that prevented him from ambulating, a balance assessment score of 4/16 and a gait assessment score of 0/12 on the Tinetti assessments, and a Barthel Index of 25/100 and a modified Rankin Scale score of 4, indicating a severe level of disability.

He exhibited severe apathy with an Apathy Evaluation Scale score of 24/36. A short NIH stroke scale assessment (NIHSS) was performed to obtain a comprehensive neurological baseline for comparison purposes and yielded a score of 4, solely for the presence of visual field and gaze deficits. Considering the long duration of his progressive decline in function, coupled with his widespread bilateral frontal lobe dysfunction, we considered a number of possible diagnoses. We believed that the most likely diagnosis was a large, slowly growing anterior cranial fossa mass lesion, such as an olfactory groove or planum sphenoidale meningioma. However, the presence of a triad of gait apraxia, cognitive decline, and urinary incontinence raised the possibility of idiopathic normal pressure hydrocephalus (NPH), although it was recognized that NPH can present with a similar clinical picture to an anterior cranial fossa mass lesion.

Other possible causes of his presentation were carefully considered, including a frontal lobe glioma, but we found it unlikely because there were no reports of headaches or seizures. Although a pituitary macroadenoma with significant suprasellar extension was a possibility given his bilateral visual loss, we were hesitant to consider this option because of the preponderance of frontal symptoms. Behavioral variant frontotemporal dementia (bvFTD) was also considered as a potential cause of his apathy and disinhibition, but we felt that his rapid decline and marked early gait disturbance made bvFTD an unlikely cause.

We systematically tested for other possible causes of his presentation, including inflammatory and metabolic processes, such as neurosarcoidosis, autoimmune encephalitis, and a hepatic encephalopathy related to his chronic liver disease. His evaluation did not demonstrate evidence of systemic inflammation or metabolic disturbances that could have explained his degree of neurological impairment.

### 2.2. Perioperative Optimization

In addition to his neurological presentation, the patient’s medical history posed additional challenges to any planned neurosurgical intervention. Specifically, he had chronic obstructive pulmonary disease (COPD) with severe obstruction (pre-bronchodilator FEV_1_/FVC ratio = 0.58), cirrhosis classified as Child–Pugh Class A, and ischemic cardiomyopathy with a severely decreased left ventricular ejection fraction (LVEF) of 40%.

These three conditions necessitated an expedited and coordinated approach to stabilize the patient’s physiology prior to surgery. The patient’s cardiology team implemented a hemodynamic management strategy using low-dose beta blockade (bisoprolol 2.5 mg/day) to reduce myocardial oxygen demand, and they carefully titrated diuretic therapy (furosemide 20 mg/day) under close surveillance of central venous pressure (CVP) measurements (kept within a range of 8–10 cm H_2_O) to maintain euvolemia without inducing acute kidney injury (his serum creatinine remained stable at 1.2 mg/dL).

In conjunction with the patient’s cardiology team, the patient’s pulmonary team developed a prehabilitation program designed to improve his lung function prior to surgery. This program included inspiratory muscle training combined with twice daily inhalation of a combination of bronchodilators (tiotropium 18 μg/day and budesonide/formoterol 160/4.5 μg BID) which significantly improved his forced expiratory volume in one second (FEV1) from a critically low percentage of predicted value (48%) to a more acceptable percentage of predicted value (55%) over a period of 5 days. Concurrently, the patient’s hepatology team identified the risks of coagulopathy associated with his cirrhosis and initiated a daily regimen of vitamin K (10 mg IV) and maintained a supply of fresh frozen plasma available for use if needed. Additionally, the hepatology team led a comprehensive review of the patient’s medications to identify those that could potentially exacerbate his hepatic dysfunction.

### 2.3. Clinical Examination and Admission Laboratory Findings

At the time of his hospital admission, the patient’s neurological examination was notable for select areas of cortical dysfunction. He was awake, but was disoriented, and had a Glasgow Coma Scale (GCS) of 14 (E4, V4, M6). His temporal and spatial orientation were grossly impaired. His visual pathway dysfunction was evident on examination, with light perception limited to the right eye and hand motion detection limited to one meter in the left eye. Both pupils reacted to light in a symmetrical and sluggish manner, suggesting pre-chiasmal compression of the optic tracts. In direct contrast to the profound dysfunction seen in his visual and spatial processing abilities, his motor system appeared to be intact. All four extremities had full power (Medical Research Council [MRC] grade 5/5), his deep tendon reflexes were normal (2+), and both lower extremities had flexor plantar responses, suggesting that the primary motor cortices were spared. However, when the patient attempted to stand, he was unable to do so because of a profound frontal lobe disequilibrium. His posture was fixed in a wide-based, shuffling position of retropulsion and motor arrest.

Laboratory studies carried out at the time of his admission provided a biochemical description of his physiological stress response. His white blood cell count was elevated (14.8 × 10^9^/L), with a predominance of neutrophils (82%), and his C-reactive protein was elevated (18 mg/L), suggesting a paraneoplastic inflammatory response. His sodium levels were diluted (131 mmol/L), and he had moderately elevated total bilirubin (1.4 mg/dL), consistent with his cirrhotic state. His albumin levels were slightly decreased (3.5 g/dL), and he had a mildly elevated NT-proBNP (580 pg/mL) consistent with his cardiomyopathy. His high sensitivity troponin I was normal (<17 ng/L). The interrelationship of the clinical findings, his underlying medical condition, and his laboratory values created an urgent need for diagnostic imaging to determine whether he had a resectable mass lesion or some other pathology.

### 2.4. Preoperative Imaging and Quantitative Metrics

Imaging studies were therefore carried out to provide anatomic information about the nature of the patient’s cerebral pathology. High-resolution MRI images with a gadolinium-based contrast agent provided an excellent anatomic depiction of the patient’s anterior cranial fossa pathology. The imaging studies depicted a large olfactory groove meningioma, which had a complex architecture requiring a precise surgical technique. The imaging studies depicted not just a mass lesion but a highly organized pathological environment that had altered the anatomic relationship of the anterior skull base structures.

As described in detail in Figure 1, the meningioma had two distinct lobes that grew at different rates, with the right lobe growing approximately 23% faster than the left lobe as determined by volumetric analysis. The sagittal plane images showed the antero-posterior diameter of the tumor to be 45.2 mm and demonstrated that the tumor covered the entire planum sphenoidale. The dural surface of the tumor contacted the skull base for 42.7 mm. Analysis of the tumor–brain interface revealed a ratio of 0.83, indicating minimal invasion of the surrounding brain tissue despite the fact that the tumor compressed the brain tissue extensively.

Utilizing complementary MRI techniques including three-dimensional volumetric image reconstruction, MR angiography, and enhanced imaging sequences (Figure 2), the authors of this study used protocol-driven quantitative analysis to evaluate both the critical neurovascular relationship to the surgical target and the quantitative characteristics of that surgical target itself. These quantitative metrics were derived directly from the native DICOM data set using a DICOM-compatible imaging workstation with multi-planar reconstruction capabilities, manual ROI (region of interest) tracing (for example, 3D Slicer, OsiriX/Horos, or another DICOM-compatible workstation), and manually drawn ROIs (calibrated angle/distance tool). In order to limit the potential for obliquity errors and improve the reproducibility of measurements, all images were rigidly realigned to a skull base standardized coordinate system (the sagittal midplane is aligned to the planum sphenoidale, the coronal plane is perpendicular to the planum sphenoidale, and the oblique parasagittal planes are parallel to each pre-chiasmatic optic nerve). MR angiography revealed that the majority of the blood supply to the tumor arose from the ethmoidal arteries (62% from the right and 38% from the left) and that these vessels converged at a point 16.3 mm posterior to the crista galli. The A2 segments of the anterior cerebral artery complex underwent significant deformation due to adaptation to the presence of the tumor. Specifically, the A2 segments were displaced upward by approximately 18.7 mm, thereby limiting the amount of space available to the surgeon (i.e., the width of the corridor between the arterial wall and the tumor capsule) to 8.3 mm. Quantification of distortion of the optic pathways was performed using a compartmental gradient method: the cross-sectional areas (CSAs) of the optic nerves were measured on coronally oriented multi-planar reconstructed (MPR) DICOM slices that were perpendicular to the direction of the optic nerves at two predefined locations along the length of the optic nerves (i.e., at the level of the optic canal opening and at the level immediately anterior to the chiasm). The measurement of the CSA of the optic nerves was performed manually using ROI tracing and did not include the area occupied by CSF in the subarachnoid space surrounding the optic nerves. This resulted in the greatest degree of narrowing occurring at the optic canal openings (a 71% reduction in CSA) followed by a relatively greater degree of tapering toward the chiasm (with a minimum of 43% CSA reduction). Finally, the angulation of the optic nerves (47.3°) was measured on the oblique parasagittal plane as the acute angle formed between an optic nerve centerline (defined as the two-point axis extending from the optic canal opening to the location immediately anterior to the chiasm) and a planum-parallel skull base reference line. Tethering of the chiasm was evaluated using a traction-stretch index (TSI), which was defined on the mid-sagittal plane as the normalized distance from the limbus sphenoidale (the posterior edge of the planum at the level of the tuberculum sellae) to the inferior midpoint of the optic chiasm. The linear distance “D” from the limbus sphenoidale to the inferior midpoint of the optic chiasm increased from a reference value of D_0_ = 8.9 mm to 17.2 mm, resulting in a TSI of D/D_0_ = 1.93. The measurements were made in the predetermined planes, repeated at least twice to decrease slice selection and partial volume artifacts, averaged for reporting, and resolved through consensus review when minor discrepancies existed.

These morphometric data were synthesized into a surgical imperative. The preoperative estimation of the optic nerve angulation (47.3°) and contact surface (210°) designed a surgical approach that had to prioritize anatomical preservation above all else. The measured 18.7 mm feature in A2 elevation and 8.3 mm reconciliation corridor suggested that a bisfrontal approach should be explored to maximize visualization whilst respecting the delicate anatomical ‘landmines’ revealed in these exhaustive imaging studies.

### 2.5. Surgical Procedure

The planning of the craniotomy began with the consideration of millimeter-level anatomic relations to the brain. The head is placed in Mayfield fixation at twenty degrees of elevation with exactly fifteen degrees of extension to place the nasion-inion vector parallel to the floor. This angulation places the orbital roofs at the highest point in the surgical field, a desired position to assist in directing the later basal dissection with gravity and also to allow the frontal lobes to relax postcisternotomy. A bicoronal incision is placed 1 cm posterior to the hairline, with subgaleal dissection in a strictly subperiosteal plane anterior to the incision. The supraorbital neurovascular bundles are noted to exit through their bony notches (true foramina in only 28% of cases) and here preserved to avoid compromising the vascularization of the pericranial graft. The temporalis fascia is incised separately in the midline, 1 cm superior to the zygomatic arch, and is protected by remaining (for the most part) in the sub-superficial musculoaponeurotic system plane, to preserve the frontal branches of the facial nerve. The craniotomy is performed utilizing five burr-holes. Two at the classic keyhole positions, five mm posterior to the frontozygomatic suture and ten mm superior to the rim of the orbit, are joined to a burr-hole at the glabella. Another two holes, paramedian, are placed 3 cm posterior to the nasion on either side of the superior sagittal sinus. Finally, the craniotome joins these points while paying particular attention to the transition from the flat to the curved surface that occurs at the level of the orbital roofs. Each time the craniotome footplate enters at a 45-degree angle, its planing proceeds into the curve to within 2 mm of the orbital roof without breaking through. The key technical step per se was the microsurgical dissection of the superior sagittal sinus from the inner table by means of a Penfield #1 dissector in a planet-to-imision sweeping motion, followed by sharp division of the dural adhesions pertinent to the sinus groove at its base with tenotomy scissors. En bloc elevation of the flap was attained without surrendering access to the sinus even with the significant frontal lobe edema present.

The opening of the dura revealed the following: the olfactory grooves were completely obliterated by the tumor; the planum sphenoidale had a hypervascular dural attachment extending 42.7 mm anteroposteriorly. The initial maneuver was one of anterior fossa devascularization: the dural attachment along the cribriform plate was coagulated with bipolar forceps (15-watt setting with continuous irrigation) severing the anterior ethmoidal arteries at their specific points of entry 16.3 mm posterior to the crista galli. The olfactory tracts were sharply divided at the cribriform plate using curved microscissors, establishing proximal control, yet accepting the sacrifice of the sense of smell.

Intracapsular decompression commenced through a cruciate capsular incision made with the tip of an arachnoid knife. Engaged at 40% fragmentation/70% aspiration was the ultrasonic aspirator in its most difficult passage. The firm, psammomatous tissue yielded, but had foci of calcific hardness, inviting an obligatory, albeit intermittent increase in amplitude to 50%. The debulking was in the radial plane, although particular attention was paid to the posterior quadrants where the tumor capsule lay adherent to the optic apparatus. Aspirations continued until precisely 70% by volume had been removed; this was the approximate volume reduction felt to be necessary. In any event, moving beyond this point into greater debulking, sure capsular handling could not be undertaken without the risk of too much traction on adjacent vital supporting tissues.

Attention was now given to the remaining neurovascular composition at the component 25×. The right optic nerve was visualized, compressed to 0.8 mm thickness with complete effacement of the pial vascularity, using a rigid no-touch technique, as microdissectors were used to develop the arachnoid plane. Starting from the optic canal, the arachnoid was developed toward the chiasm, recognizing the fine pial vessels responsible for the postoperative recovery of vision. The internal carotid artery was circumferentially dissected in its supraclinoid segment with the identification of the “hooked” vessels, superior hypophyseal arteries coursing medially to the optic apparatus. The requisite extreme caution was taken in dissecting the anterior cerebral artery complex, with particular reference to the many branching perforating vessels to the hypothalamus and optic apparatus, all of which measured 0.2–0.5 mm in diameter at this stage. All gross traction on any of these vessels led to avulsion. The identical sequence was then carried out on the contralateral side. Expectantly the synchronous visual fields of each eye returned together, and unexpectedly dense arachnoid adhesions were found between the tumor capsule and the left optic nerve, mandating sharp dissection with micro-scissors rather than blunt dissection. The final detachment was accomplished with delicate coagulation of the dural attachment to the tuberculum sellae with preservation of the entire pituitary stalk characterized by a vascular stripe and by a striking tan coloration but also with care to preserve the stalk’s portal venous system. Closure was carried out by multilayer reconstruction whereby the cranialized frontal sinus was sealed with the vascularized pericranial flap and watertight dural closure using a running 4-0 sheer Nurolon suture in continuous locking fashion was performed. The bone flap was rigidly secured with miniscipes of titanium at six points to prevent sinking postoperatively. Postoperative imaging confirmed total resection and optic apparatus decompression. The patient had improved vision acuity, and they did not experience frontal lobe symptoms. The technical nuances—from the angles to choose from to approach the pathology to protecting the vasculature at 1 mm intervals—were contributory.

### 2.6. Postoperative Outcomes and Follow-Up

The patient demonstrated gradual neurological improvement in all domains compromised by the tumor. Within the first week postoperatively, overtly disordered frontal functioning, which had defined much of his clinical picture, began to show definite recovery. He became less bedbound, sitting with minimal assistance on postoperative day 4 and standing with some support by the end of week. His overt motor symptomatology, with its most conspicuous feature being the magnetic apraxia, also markedly improved. Cognitively there were demonstrable parallel gains. His MoCA score improved from 12/30 (preoperative) to 18/30, with demonstrable greater performance with attention, working memory, and executive function tasks. His Front Assessment Battery score went from 8/18 to 12/18—a reflection of recovering function in the cortico-cortical fraction junctions. His vision improved in advance of optical rehabilitation, from light perception to counting fingers at a meter from his right finger and from hand motion to counting fingers at three meters on his left. These changes coincided reassuringly with evidence of the ocular decompression of the pre-chiasmatic optic apparatus. The immediate postoperative films, shown in Figure 3, confirmed our desired goals. The one-week CT confirmed the complete resection of tumor mass with the expected postoperative anatomy, small amounts of frontal pneumocephalus (approx. 15 cc volume), the expected linear enhancement in the surgical bed, and complete decompression of the frontal lobes, with re-expansion and resolution beginning in the basal cisterns and a definite release of optic apparatus compression. Given the complexity of the patient’s comorbidities, this was an accomplishment. He remained volume-contracted but was stable on diuretic therapy. His cardiac status remained stable on diuretics titrated upwards to maintain euvolemic status without renal compromise (his creatinine remained stable at 1.3 mg/dL). Bronchodilator therapy was restarted to maintain his pulmonary status, hemodynamic monitoring was pursued, and close attention was paid to his coagulopathy. There were no hemorrhagic complications, and the remainder of his tablets were restarted (his corticosteroids, proton-pump inhibitors, and antiepileptic prophylaxis for 2 weeks).

Follow-up at six months showed him transformed. His functional status, as measured by the Barthel Index, improved from 25/100 to 75/100. His mRS likewise improved from 4 (severely disabled) to 2 (slight disability). Cognitive testing revealed a score of 24/30 on the Montreal Cognitive Assessment with an improved Frontal Assessment Battery of 15/18. His vision was improved to 20/100 right eye and 20/80 left eye, with correction and formal visual field testing showing a dramatic improvement from the hands to the face in the severe preoperative follow-up testing. The 6-month follow-up also showed durable resection. These findings confirm stable decompression without any new findings from the 23 immediate postoperative films (Figure 4).

The long-term clinical outcome, including clinical and serial radiographic assessments, is a positive one for this patient after undergoing a complex cranial base operation. In addition to documenting a favorable and prolonged trend of recovery, this assessment also documents measurable and meaningful improvement in all previously affected neurological functions and confirms the technical success and stability of the initial surgical intervention via both short- and long-term assessments.

This clinical report describes the treatment of a complex neurosurgical case of a giant olfactory groove meningioma that was present in an elderly individual who had multiple organ system comorbidities. This case is illustrative of three clinically significant challenges: the interpretation of progressive frontal lobe syndrome; the selection of a surgical approach for the treatment of large anterior skull base lesions; and the perioperative care of patients who have compromised cardiovascular, pulmonary, and hepatic function. The documented improvement in this patient’s neurological status, specifically regarding vision and cognition, as well as the demonstrated radiographic evidence of successful decompression of the tumor, could be helpful in providing an understanding of possible opportunities for functional recovery in advanced presentations.

The systematic approach to the management of this patient, from preoperative optimization through the conduct of the surgical intervention and the implementation of postoperative rehabilitation, may provide a useful paradigm for the management of similar complex cranial base pathologies. Therefore, the clinical information provided in this case will contribute to the body of knowledge related to the management of neurosurgical patients where the pathological complexity of the lesion and the extent of the medical comorbidities are significant.

## 3. Discussion

### 3.1. Case-Specific Rationale and Surgical Strategy

In this specific case, the most important aspect of skull base surgery in a medically fragile elderly individual is preserving function using a structured plan prior to surgery and reducing risks from multiple disciplines, instead of focusing on radicalism as the sole goal of surgery. The use of a bifrontal corridor in the current case was based on the large size of the tumor (approximately 81 cm^3^), the tumor’s extension bilaterally past the middle visual corridor, and its relationship with the artery of cerebral circulation and optic apparatus. All of the abovementioned factors lead to using a midline surgical exposure to allow for optimal viewing of the optic–carotid complex and to systematically de-vascularize and internally decompress the capsule before it is mobilized; all of which are principles previously published regarding the selection of approaches in the treatment of large olfactory groove meningiomas [7].

The main technical concepts used in the operation included staging the internal decompression to minimize wall tension within the capsule and reduce traction on the optic apparatus and the perforator rich area surrounding the artery of cerebral circulation and dissecting the arachnoid plane under magnified conditions while preserving the micro vessels of the pia mater. While these concepts are well-known to surgeons operating in the skull base, they become clinically significant when treating elderly individuals, since the physiological reserves are decreased and even minimal new deficits could potentially affect the ability of the patient to be independent postoperatively.

### 3.2. Contribution of Quantitative Indices to Operative Decision-Making

One major purpose of this study was to demonstrate how anatomical quantifiable indices can improve the planning of surgery in the treatment of giant anterior skull base tumors. Of particular interest were two measurements: optic pathway angulation and the TSI. As opposed to being predictive measures, these measurements are presented as definitions of the relevant anatomical structures relative to the degree of risk in each structure. They provide a method for translating the visually apparent deformations in anatomical structures to a number that can be compared between different times or among different groups in future studies. For this patient, the combination of the severe deformation of the optic apparatus including narrowing of the optic canal and a significantly elevated angulation signal combined with a narrow artery of cerebral circulation–tumor corridor provided a measurement-based rationale to select a surgical corridor that optimizes bilateral visualization and minimizes “blind” dissection of critical perforating vessels around them.

It is also important to recognize the clinical utility of the indices discussed above in hypothesis generating at the single-patient level. The potential immediate applications of these indices may be to standardize reports of surgical outcomes, to enhance communication among surgeons, and to enable informed preoperative counseling regarding risks associated with surgery in cases where qualitative descriptions (“severe,” “significant,” “encased”) are difficult to quantify. In Table 1, we have summarized the various levels of factors that will determine the successful outcome of surgery and long-term control of giant olfactory groove meningiomas.

### 3.3. Geriatric Risk Stratification as a Parallel Framework

To address the problem of improving neurosurgical care for elderly individuals, researchers have developed other methods of quantifying an individual’s potential to recover or withstand treatment by creating predictive indexes (scores). For instance, in high-grade gliomas, Bianconi et al. introduced the DAk-75 index system to assist in determining whether a patient aged 75 or greater should undergo surgery on the basis of their preoperative risk factors and the risk of mortality at one year postoperatively [24].

Although the biological features of meningiomas and the goals for treating these tumors differ significantly from those of malignant gliomas, the overall strategy of replacing clinical judgment with standardized, transparent categorization is similar. Both strategies aim to provide context for the structural capacity of a patient to withstand aggressive treatment (in terms of potential for survival), as well as for anatomical hazards and technological challenges associated with specific interventions. Ultimately, combining these two levels of risk assessment—patient-specific fragility and anatomical-specific risk—may yield a comprehensive decision-making model for geriatric neurosurgical practice [25].

### 3.4. Therapeutic Alternatives and Adjuvant Considerations in Context

While minimally invasive endoscopic endonasal techniques are capable of effectively addressing selected lesions in the midline skull base, the management of large olfactory groove meningiomas remains challenging due to the extensive lateral extent of these tumors, optic apparatus compromise, and/or complex vascular relations. While the benefits of less cranial incision-related morbidity may outweigh the limits in lateral control and vascular dissection, as well as reported CSF leakage rates, in these situations [26], radiotherapeutic strategies will continue to play an important role in managing residual or recurrent WHO Grade I meningiomas and in patients who are not candidates for surgical treatment based upon their level of surgical risk. Stereotactically delivered radiation, either conventionally fractionated or hypofractionated, allows for durable local tumor control while adhering to dose constraints imposed by the optic pathways. Proton therapy may allow for a decreased total body dose in the presence of irregular skull base geometry; however, longer-term follow-up studies remain relatively underdeveloped [27]. When used in conjunction with an understanding of the patient’s individual situation, these options establish what is feasible to accomplish if it is not possible to achieve safe margins of resection or completely decompress the tumor.

### 3.5. Future Directions: Technologies That Could Refine and Validate the Proposed Metrics

Technologies that enhance operative capabilities and imaging will probably be most beneficial when they reinforce the accuracy of the measurements involved in quantitative planning. Enhanced reality navigation systems and more sophisticated registration protocols will probably lower the degree of uncertainty associated with translating preoperatively derived morphometric data into the intraoperative environment, particularly in areas such as the optic canals and ACA perforator zones [28]. Robotic or tremor-filtered instrumentation may also facilitate stable performance during microsurgical execution within millimeter-scale dissecting corridors, thereby possibly reducing the degree of variation among surgeons regarding how “atraumatic” capsular mobilization is performed [29]. Quantitative imaging biomarkers—including radiomics and diffusion models—may also help to differentiate reversible from irreversible damage to neural pathways, providing a more direct link between the preoperative indices (for example, angulation/TSI and CSA gradients at the canal level) and postoperative recovery trajectories [30]. Thus, emerging technologies will not represent an additional layer of narrative but rather tools that may (i) increase the reliability of measurements, (ii) permit multi-center calibration, and (iii) elucidate how deformations in anatomic morphology relate to clinically relevant outcomes [31].

### 3.6. Limitations and Validation Priorities

Published large series of olfactory groove meningiomas that focus on “large” (greater than 6 cm) lesions and their subsets have emphasized the same two clinical themes: first, patients with olfactory groove meningiomas commonly present late with a significant combination of frontal lobe dysfunction and visual impairment, and second, the risk of surgery is based on a limited number of anatomic limitations related to individual patient anatomy (anatomical distortion of the optic nerve and chiasma, location/proximity of the ACA, and anatomical pathway dimensions) [32]. Furthermore, data from the long-term follow-up of large groups of patients with giant OGMs suggest that meaningful recovery of cognitive behavior and vision can be achieved postoperatively (after decompression) when using a treatment strategy aimed at achieving the maximum amount of tumor removal as safely as possible while preserving all critical neurovascular structures [33].

However, the numerous variations in surgical strategies used in reported series also highlight the continued need for the development of standardized and consistent methods of describing optic nerve/chiasmal deformity and vascular deformities that will allow for better standardization of preoperative planning and counseling, as well as facilitating comparison among studies [34].

A major benefit of this study lies in its ability to convert the clinical deformities of the optic pathway into quantitative measures (optical angulation and the traction-stretch index) that are easy to reproduce and measure using a transparent process. The results from this study also provide multiple domains of objective outcomes (vision; cognition/executives; gait/functioning) at two time points (early and mid-term) after surgery. This may indicate that quantitative anatomical information can be incorporated into surgical planning for each patient prior to surgery without altering the basic principles of microsurgery.

The primary limitation of this study is that the proposed indices were not designed to serve as independent predictive tools. Rather, their immediate utility exists within the context of practice: providing a common language to describe complex anatomical relationships, facilitating the exchange of ideas among surgeons, and serving as an anchor point for discussions regarding counseling and operative approach selection when terms like “severe compression” and/or “tight corridor” are often subjectively defined and/or unmeasurable.

As such, the next logical step in research would be to assess the inter-rater reliability of the proposed measures on different imaging platforms, test the robustness of the measurements to variations in the imaging planes selected and the effects of partial volume artifacts, and determine if a relationship exists between the proposed measures and standardized postoperative changes in vision and cognition among larger groups of patients.

## 4. Conclusions

A model that illustrates how microsurgical technique and a perioperative optimization plan may be used in conjunction with anatomical quantification of patient-specific anatomy is presented in Contemporary Skull Base Surgery. A structured multidisciplinary evaluation of each patient’s co-morbidities and an anatomically defined analysis of the individual’s morphology prior to surgery can provide a useful paradigm for similar, high-risk cranial base surgeries, particularly for those which are limited by distortion of the optic apparatus or complex neuro-vascular relationships, and therefore the surgical field.

The proposed measures (optic angulation and the traction-stretch index) should be viewed at this time as anatomically based definitions of anatomy rather than predictive values and must be validated in a large number of centers utilizing standardized outcome assessments of both visual function and cognitive status and also standardized assessments of intra-rater reliability. Additionally, advancements in the use of computational anatomy, molecular profiling, and perioperative protocols will allow for improved estimation of risk and increased likelihood of safe tumor resection in patients with large skull-based tumors and who are considered to be medically complex.

## Figures and Tables

**Figure 1 diagnostics-16-00127-f001:**
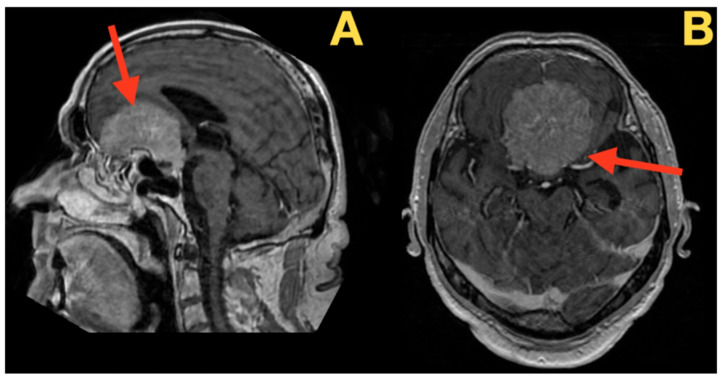
Preoperative high-resolution MRI with gadolinium contrast. (**A**) Sagittal T1-weighted image: This image illustrates a large tumor in the anterior cranial fossa (red arrow), which has a wide dural base extending from the crista galli to the tuberculum sellae. The tumor is causing significant upward displacement and compression of both frontal lobes and is completely obliterating the olfactory sulci. There is a posterior displacement of 18.7 mm of the frontal horn. The optic chiasm is also elevated and compressed against the planum with a 47-degree change in angulation and the pituitary stalk is being displaced posteriorly but can still be identified. The tumor extends posteriorly to the limb sphenoidale with a 3.2 mm cerebrospinal fluid cleft preservation anterior to the brain stem. (**B**) Coronal T1-weighted view: this view further illustrates the tumor’s bilateral extension (red arrow) along with the asymmetric compression of the orbital gyri with greater compression on the right by 4.3 mm. Both optic nerves have significantly reduced cross-sectional areas with the right optic nerve reduced to 27% of normal and the left optic nerve reduced to 34% of normal. The A2 segments of the ACAs are elevated and partially encircled by the tumor capsule with a 210-degree contact surface. The supraclinoid internal carotid arteries are laterally displaced by a mean of 4.0 mm; however, they remain separated by clear arachnoid planes and preserve their flow voids.

**Figure 2 diagnostics-16-00127-f002:**
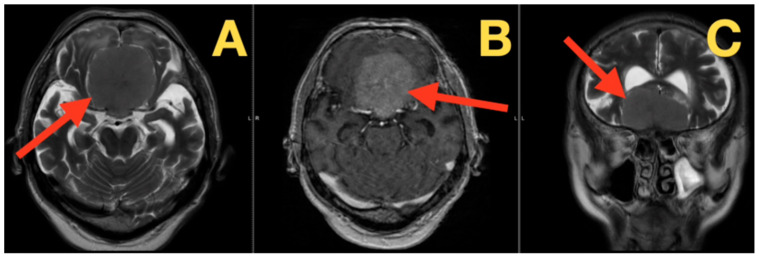
Preoperative MRI with 3D reconstruction and quantitative volumetrics. (**A**) Three-dimensional volume rendering (superior view): three-dimensional measurement of tumor size (68.3 mm × 52.1 mm × 45.2 mm) with calculation of sphericity index (0.78) and volume (81.2 cm^3^). The reconstruction (red arrow) clearly demonstrates that the tumor completely covers the cribriform plate and has an anterior extension of 7.3 mm beyond the crista galli and a posterior extension of 4.8 mm beyond the tuberculum sellae. (**B**) Sagittal T2 FLAIR reconstruction: this reconstruction clearly illustrates the relationship between the tumor and the anterior skull base, as well as measuring the planum-to-tuberculum coverage ratio (1.42:1) (red arrow). The significant compression of the frontal lobes shows a superior displacement of 28.3 mm of the rectus gyrus. The significant vasogenic edema within the bilateral frontal white matter is clearly visualized. (**C**) Axial T1-weighted with contrast (posterior–anterior angulation): this image clearly delineates the deformation of the optic apparatus, as well as calculating the optic canal invasion index (0.32 right, 0.29 left) (red arrow). The anterior cerebral artery “smile” configuration shows an elevation of 18.7 mm from the planum. The fronto-polar arteries demonstrate an inferior deflection of 43 degrees around the tumor capsule.

**Figure 3 diagnostics-16-00127-f003:**
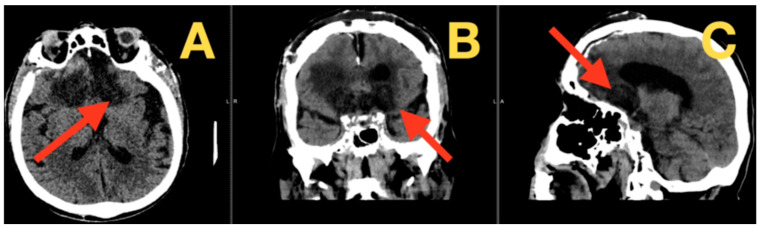
Postoperative computed tomography at one week. (**A**) Axial view: this view demonstrates the postoperative bed with anticipated postoperative enhancement and minimal residual pneumocephalus. The frontal lobes (arrow) are showing early re-expansion of the frontal lobes with the resolution of the mass effect and normalization of the ventricular configuration. (**B**) This view confirms complete removal of the tumor and preservation of the optic apparatus, which is no longer compressed and has restored peri-optic cerebrospinal fluid spaces (arrow). (**C**) Sagittal view: this view illustrates the skull base reconstruction and the relationship of the surgical site (arrow) to the frontal lobes and residual sinus structures and maintains anatomically correct positioning.

**Figure 4 diagnostics-16-00127-f004:**
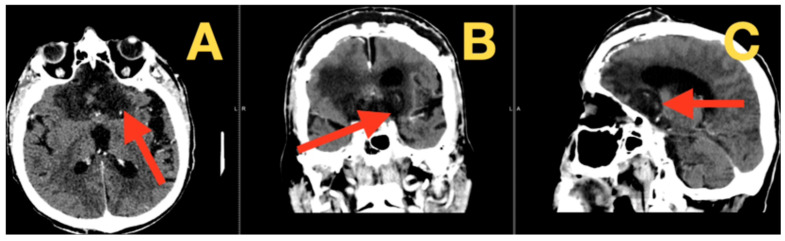
Six-month follow-up computed tomography. (**A**) Axial view: this view demonstrates complete resolution of postoperative changes with normal cerebral anatomy (arrow) and stable surgical site. (**B**) Coronal view: this view confirms continued decompression of the optic apparatus and frontal lobes with normal basal cisterns (arrow). (C) Sagittal view: this view further confirms stable postoperative anatomy without recurrent lesion (arrow), with preserved midline CSF pathways and absence of hydrocephalus.

**Table 1 diagnostics-16-00127-t001:** Each row maps a specific decision or control layer to its actionable technical or biological lever, the quantitative target defining optimal execution, the principal risk being constrained, and the outcome-relevant metric that operationalizes success.

Decision/Control Layer	Key Technical or Biological Lever	Operational Thresholds/Quantitative Targets	Primary Risk Mitigated	Outcome-Relevant Signal/Metric	References
Approach Selection Logic	Corridor choice based on morphometrics and vascular topology	Volume >80 cm^3^; bilateral reach beyond mid-pupillary lines; ACA encasement severity guiding midline exposure need	Incomplete resection from limited lateral/vascular access	Higher gross-total resection probability by maximizing opticocarotid visualization	[8,9]
Debulking Phase Control	Ultrasonic aspiration and staged internal decompression	Aspirator amplitude 40–50%; irrigation pressure ~5 mmHg; debulking to 65–75% volume reduction before capsular work	Thermal or traction injury; unsafe capsular mobilization	Safe transition to capsule dissection with reduced wall tension	[10]
Optic Apparatus Microdissection	High-magnification pial-preserving peel	25× magnification; intermittent 30 °C irrigation; dissection along pial vascular planes	Ischemic optic neuropathy; pial vessel avulsion	Preservation of microvascular perfusion despite extreme preop compression	[11]
Intraoperative Neurophysiology Guardrails	Continuous sensory/brainstem monitoring as live safety envelope	BAEP latency drift ≤0.8 ms; SSEP amplitude ≥50% baseline	Silent brainstem/tract injury during deep manipulation	Real-time functional stability predicting favorable neurologic outcome	[12]
Comparator Technique Boundary Conditions	Endonasal/endoscopic limits in giant lesions	Lateral reach limit ~30° from midline; CSF leak 12–18% vs. 3–5% transcranial; vascular encasement handling constraints	Inadequate lateral control; high leak burden; vascular compromise	Defines selection ceiling for minimally invasive routes	[13]
Adjuvant Radiotherapeutic Optimization	Fractionated vs. hypofractionated stereotactic dosing	FSRT: 5-year PFS ~95% (WHO I); HFRT: 25–30 Gy/5 fx, local control 88–92%; optic dose cap 8–10 Gy	Recurrence progression while protecting optic pathways	Durable control in residual/recurrent disease with optic safety	[14]
Proton Dosimetry Advantage Window	Integral dose reduction for complex skull-base geometry	Integral brain dose ↓ 50–60% relative to photons	Cognitive/vascular late-effects from unnecessary dose	Preferential for irregular targets near eloquent structures	[15]
Edema and Seizure Pharmacoprotection	Steroid taper + brief antiepileptic prophylaxis	Dexamethasone 16 mg/day, taper 14 days; levetiracetam 500 mg BID ×7 days	Edema-driven neurological decline; periop seizures; extended AED cognitive burden	Balanced physiological stabilization with minimized adverse profile	[16]
Stress Ulcer and Systemic Safety Net	GI prophylaxis allied to prolonged cranial surgery	Pantoprazole coverage for high-stress surgical physiology	Hemorrhagic GI complications	Protection against stress ulcer incidence in extended procedures	[17]
Hepatic Risk Coagulation Governance	TEG-guided transfusion and conservative plasma use	Platelets >50 k/µL; fibrinogen >150 mg/dL; FFP only if INR >1.8	Venous bleeding or overcorrection thrombosis in liver impairment	Hemostasis precision under non-standard coagulation biology	[18]
Cardiopulmonary Equilibrium Maintenance	Tight BP/HR variability control to preserve perfusion	SBP within ±20% baseline; HR variability within ~15% optimal band	Hypoperfusion, hyperemia, or cardiac destabilization	Stable cerebral perfusion across long operative periods	[19]
Genotype-Linked Behavior Forecasting	Driver mutations and epigenetic subtyping	AKT1(E17K) 8–15%; NF2 loss 40–60%; methylation classes 6 groups; H3K27me3 loss and TERT-p mutations 6–8%	Underestimated recurrence risk in “benign” histology	Molecular stratification refines recurrence surveillance intensity	[20]
Liquid Biopsy Signal Layer	CSF vs. plasma ctDNA sensitivity windows	CSF tumor DNA sensitivity ~89%; plasma ctDNA recurrence signal ~73%, lead time 6–12 mo	Delayed detection of molecular recurrence	Early, minimally invasive relapse forecasting	[21]
Technique-Bound Risk Envelope	Exposure benefit traded for approach-specific morbidity	Anosmia~100%; frontal retraction injury 8–12%; SSS sacrifice-related venous infarct 5–8%	Functional loss and venous complications intrinsic to wide corridors	Defines consent-critical risk profile of bifrontal route	[22]
Limitations/Validation Horizon	Novel indices need cohort-scale confirmation	Optic angulation 47.3°; traction-stretch index 1.93; validation target >200 pts	Premature generalization of single-center metrics	Identifies parameters requiring multi-center calibration	[23]

**Note:** **>** indicates greater than; **≤** indicates less than or equal to; **~** indicates approximately; **°** indicates degrees; **×** indicates “for” (duration) or multiplied by; **↓** indicates relative decrease; **±** indicates variation around baseline; **%** indicates percent. Units/abbreviations: **cm^3^**, cubic centimeters; **mmHg**, millimeters of mercury; **mg/day**, milligrams per day; **Gy**, gray; **fx**, fractions; **BID**, twice daily; **k/µL**, thousands per microliter.

## Data Availability

The data presented in this study are available on request from the corresponding author due to ethical and privacy constraints associated with patient-specific data.

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
