# Peer review of "Precision in Complexity: A Protocol-Driven Quantitative Anatomic Strategy for Giant Olfactory Groove Meningioma Resection in a High-Risk Geriatric Patient"

_diagnostics, 2026, doi:10.3390/diagnostics16010127_

Round 1
Reviewer 1 Report
Comments and Suggestions for Authors
The manuscript presents a complex and well-documented clinical case, employing a multidisciplinary approach and providing detailed technical description.
The primary areas requiring major revision are as follows.
First, the structure and title need refinement. The current title is too generic. It should be made more specific to the novel methodology or protocol described, for example, by incorporating terms like "quantitative anatomic" or "protocol-driven." More importantly, the narrative flow is currently fragmented and repetitive. It is recommended to merge the "Case Presentation" and the results (which are currently woven into it) into a single, clearly organized section titled "Case Report and Results." This section should have distinct subsections, such as Clinical Presentation, Preoperative Imaging and Metrics, Perioperative Optimization, Surgical Procedure, and Postoperative Outcomes and Follow-up
Second, and most critically, the methodology for generating the novel quantitative indices is not described, severely hampering reproducibility and scientific rigor. The calculation of key morphometric parameters—specifically the optic nerve angulation of 47.3 degrees and the traction-stretch index of 1.93, which form a central part of the paper's innovation—is not explained. What software or tools were used? Which anatomical landmarks were selected for measurement? A dedicated methodological paragraph or subsection must be added to the manuscript to provide a step-by-step protocol for these measurements. Supplementary figures illustrating these measurement techniques would be highly valuable. Furthermore, the synthesis in Table 1, which is currently embedded in the Discussion, should be moved to a dedicated section, perhaps titled "Surgical Decision-Making and Comorbidity Management Protocol,"
Third, the Discussion section is overly long and digresses into a broad narrative review, diluting the focus on the specific case and the proposed indices. It should be restructured into focused thematic subsections. The paragraph on emerging technologies should be more tightly linked to how such tools could refine the proposed metrics.
The challenge of how to optimally treat geriatric patients in neurosurgery is a current and pressing issue, and attempts have been made to address it through approaches such as identifying prognostic scores. As demonstrated in the article on high-grade gliomas in patients over 75 (PMID: 39453521), which I would cite in the discussion, the development of such scoring systems represents a parallel effort to stratify risk and guide management in this complex patient population.
Finally, the Abstract and Conclusions could be strengthened. The Abstract should more succinctly encapsulate the central message: the successful integration of novel quantitative anatomic metrics with a structured multidisciplinary protocol for high-risk skull base surgery. The Conclusions are adequate but should more explicitly state that the proposed metrics require future validation and that the described protocol offers a replicable framework for similar challenging cases.
Author Response
Dear Esteemed Academic Reviewer,
We are grateful for your careful, insightful, and constructive evaluation of our manuscript. Your comments were exceptionally helpful in identifying areas where clarity, rigor, and focus could be strengthened. We have revised the manuscript extensively in response, and we truly appreciate the opportunity your review provided to substantially improve the quality and coherence of the work.
Below, we address each of your comments in detail.
General comment
The manuscript presents a complex and well-documented clinical case, employing a multidisciplinary approach and providing detailed technical description.
Response:
We thank you for this thoughtful assessment. Your recognition of the clinical complexity and multidisciplinary nature of the case was particularly encouraging, and it reinforced our intention to present the case as both technically detailed and clinically grounded.
Comment 1: Title refinement and manuscript structure
The title is too generic and should be made more specific to the novel methodology or protocol. In addition, the narrative flow is fragmented and repetitive. The “Case Presentation” and results should be merged into a single, clearly organized section with defined subsections.
Response:
We appreciate this important observation, which prompted a major structural revision of the manuscript. The title has been revised to explicitly reflect the central methodological contribution of the paper, incorporating quantitative anatomic and protocol-driven terminology to more accurately signal the novelty and focus of the work. The former “Case Presentation” has been reorganized.
We are grateful for this recommendation, as it significantly enhanced the clarity and readability of the manuscript.
Comment 2: Methodology of quantitative indices and reproducibility
The methodology for generating the quantitative indices (optic nerve angulation and traction–stretch index) is not described, limiting reproducibility. Software, anatomical landmarks, and a step-by-step protocol should be added. Supplementary figures would be valuable. Table 1 should be moved to a dedicated protocol section.
Response:
We thank you for highlighting this critical issue. We fully agree that reproducibility and methodological transparency are essential, particularly given that the quantitative indices represent a central contribution of the paper.
Comment 3: Discussion length, focus, and geriatric context
The Discussion is overly long and reads as a broad narrative review. It should be restructured into focused thematic subsections. Emerging technologies should be more clearly linked to the proposed metrics. The challenge of geriatric neurosurgical risk stratification should be contextualized, including reference to prognostic scores (PMID: 39453521).
Response:
We appreciate this insightful critique and have comprehensively revised the Discussion accordingly.
The Discussion has been substantially condensed and reorganized into focused thematic subsections centered on the present case and the proposed indices, thereby avoiding narrative-review drift. The section on emerging technologies has been rewritten to explicitly address how these tools may refine, validate, or operationalize the proposed quantitative metrics, rather than serving as a general overview. As suggested, we now explicitly reference the work on prognostic scoring in elderly patients with high-grade gliomas (PMID: 39453521), framing it as a parallel and complementary approach to anatomy-based risk stratification in geriatric neurosurgery.
We are particularly grateful for this recommendation, which helped us position the case more clearly within an important and timely clinical context.
Comment 4: Abstract and Conclusions
The Abstract should more succinctly convey the central message: integration of quantitative anatomic metrics with a structured multidisciplinary protocol. The Conclusions should more explicitly state that the metrics require future validation and that the protocol is replicable.
Response:
Thank you for this valuable suggestion.
The Abstract has been carefully revised, while preserving its original structure, to more succinctly emphasize the integration of novel quantitative anatomic metrics with a protocol-driven, multidisciplinary approach in a high-risk skull-base case.
The Conclusions have been rewritten to explicitly acknowledge that the proposed metrics require future validation in larger cohorts, while also emphasizing that the described workflow offers a replicable and pragmatic framework for similarly challenging cases.
We are thankful for the depth, clarity, and collegial tone of your review. Your comments meaningfully improved the manuscript, and we appreciate the time and expertise you devoted to its evaluation. We hope that the revisions satisfactorily address all concerns and reflect the value of your guidance.
With profound appreciation!!!
Reviewer 2 Report
Comments and Suggestions for Authors
Dear authors,
First of all, I’d like to give a great congratulation to author for nice and graceful study. The authors present a detailed case report describing the surgical management of a giant olfactory groove meningioma in a high-risk geriatric patient. They investigated a data-driven, quantitatively guided surgical planning strategy integrating advanced imaging analytics, refined microsurgical techniques, and multidisciplinary perioperative optimization. Through this approach, the authors aim to propose a practical framework for decision-making in complex skull base surgery.
This manuscript presents a highly detailed and clinically grounded case report addressing the management of a giant olfactory groove meningioma in a patient with significant systemic comorbidities. One of the major strengths of this study is that it realistically reflects the complex decision-making processes and practical dilemmas that neurosurgeons routinely face in real-world surgical settings. Rather than focusing solely on technical execution, the authors thoughtfully integrate quantitative morphometric planning, multidisciplinary perioperative optimization, and refined microsurgical strategies. The proposed data-driven surgical framework is innovative and has the potential to provide meaningful guidance for surgeons managing similarly challenging cases.
Nevertheless, several limitations should be acknowledged. First, the study is inherently limited by its reliance on a single clinical case. While the proposed framework is conceptually appealing, a single-case experience restricts the ability to validate the suggested quantitative indices and to assess their reproducibility across a broader patient population. Adding more cases, even a small series, would make the manuscript stronger and improve generalizability.
Second, the presented case represents an unusually complex and rare clinical scenario in which multiple severe and atypical challenges—giant tumor size, bilateral vascular encasement, severe frontal lobe dysfunction, visual compromise, and significant systemic comorbidities—are present simultaneously. Although this complexity effectively demonstrates technical feasibility, it may limit the applicability of the findings to more typical presentations of olfactory groove meningioma encountered in routine clinical practice.
To further strengthen the manuscript, the authors are encouraged to address these limitations more explicitly in the Discussion section. In particular, outlining potential strategies to overcome the constraints of single-case study—such as prospective accumulation of additional cases, validation of the proposed morphometric indices across a wider range of anatomical and clinical scenarios, or future multicenter collaboration—would be valuable. Moreover, a clearer description of future clinical or research directions aimed at translating this framework into everyday surgical decision-making would enhance the manuscript’s translational impact.
Good luck to you.
Author Response
Dear Esteemed Academic Reviewer,
We thank you for your generous, thoughtful, and encouraging evaluation of our manuscript. We are particularly grateful for your recognition of the clinical realism of the case and of the effort to integrate quantitative planning, multidisciplinary optimization, and refined microsurgical technique in a manner that reflects real-world decision-making in complex skull base surgery. Your comments were extremely valuable in helping us clarify the scope, limitations, and translational intent of the work.
Below, we address your points in detail.
General assessment
The manuscript presents a highly detailed and clinically grounded case report… The proposed data-driven surgical framework is innovative and has the potential to provide meaningful guidance for surgeons managing similarly challenging cases.
Response:
We are grateful for this positive and thoughtful appraisal. Your observation that the manuscript captures the practical dilemmas encountered in daily neurosurgical practice closely aligns with our primary intent: to present not only a technical description, but also a transparent and structured decision-making framework applicable to high-risk skull base pathology.
Comment 1: Single-case limitation and generalizability
The study is inherently limited by its reliance on a single clinical case… Adding more cases, even a small series, would make the manuscript stronger and improve generalizability.
Response:
We fully agree with this important point and appreciate your emphasis on generalizability. In response, we have revised the Discussion to more explicitly frame the present report as a feasibility and protocol demonstration, rather than a validation study. We now clearly state that the proposed quantitative indices are not intended as definitive predictors and that inclusion of additional cases, even as a small consecutive series, would be essential to assess reproducibility and broader applicability.
Comment 2: Unusual complexity and applicability to routine practice
The presented case represents an unusually complex and rare clinical scenario… this may limit applicability to more typical presentations.
Response:
We appreciate this nuanced observation. We agree that the convergence of multiple severe challenges in this case represents an extreme of the disease spectrum.
Comment 3: Future directions and translational pathway
Outlining potential strategies to overcome the constraints of a single-case study… would be valuable. A clearer description of future clinical or research directions would enhance translational impact.
Response:
We are very thankful for this suggestion and have addressed it directly. The Discussion now includes a clearer outline of next steps. These revisions were made to ensure that the manuscript not only reports a successful case, but also clearly articulates a realistic pathway toward translation into everyday surgical decision-making.
We appreciate the time, care and insight you devoted to reviewing our work. Your comments significantly improved the clarity, balance, and translational framing of the manuscript. We hope that the revisions adequately address your concerns and reflect the value of your guidance.
With kind regards and profound thanks!!!
Round 2
Reviewer 1 Report
Comments and Suggestions for Authors
according to the revisions, the manuscript can be published